# Multi-Institutional Evaluation of Pathologists’ Assessment Compared to Immunoscore

**DOI:** 10.3390/cancers15164045

**Published:** 2023-08-10

**Authors:** Joseph Willis, Robert A. Anders, Toshihiko Torigoe, Yoshihiko Hirohashi, Carlo Bifulco, Inti Zlobec, Bernhard Mlecnik, Sandra Demaria, Won-Tak Choi, Pavel Dundr, Fabiana Tatangelo, Annabella Di Mauro, Pamela Baldin, Gabriela Bindea, Florence Marliot, Nacilla Haicheur, Tessa Fredriksen, Amos Kirilovsky, Bénédicte Buttard, Angela Vasaturo, Lucie Lafontaine, Pauline Maby, Carine El Sissy, Assia Hijazi, Amine Majdi, Christine Lagorce, Anne Berger, Marc Van den Eynde, Franck Pagès, Alessandro Lugli, Jérôme Galon

**Affiliations:** 1Department of Pathology, UH Cleveland Medical Center, Cleveland, OH 44106, USA; josephe.willis@uhhospitals.org; 2Pathology Department, John Hopkins, Baltimore, MD 21287, USA; rander54@jhmi.edu; 3Department of Pathology, Sapporo Medical University School of Medicine, Sapporo 060-8556, Japan; torigoe@sapmed.ac.jp (T.T.); hirohash@sapmed.ac.jp (Y.H.); 4Department of Pathology and Molecular Genomics, Providence Portland Medical Center, Portland, OR 97213, USA; carlo.bifulco@providence.org; 5Institute of Pathology, University of Bern, 3008 Bern, Switzerland; inti.zlobec@pathology.unibe.ch (I.Z.); alessandro.lugli@pathology.unibe.ch (A.L.); 6INSERM, Laboratory of Integrative Cancer Immunology, 75006 Paris, France; bernhard.mlecnik@crc.jussieu.fr (B.M.); gabriela.bindea@crc.jussieu.fr (G.B.); florence.marliot@aphp.fr (F.M.); nacilla.haicheur@aphp.fr (N.H.); tessa.fredriksen@crc.jussieu.fr (T.F.); amos.kirilovsky@gmail.com (A.K.); benedicte.buttard@crc.jussieu.fr (B.B.); angela.vasaturo@ultivue.com (A.V.); lucie.lafontaine@crc.jussieu.fr (L.L.); mabpau@gmail.com (P.M.); carineelsissy@hotmail.com (C.E.S.); assia.hijazi@sorbonne-universite.fr (A.H.); amine.majdi.pro@gmail.com (A.M.); christine.lagorce@aphp.fr (C.L.); aberger@ghsj.fr (A.B.); franck.pages@egp.aphp.fr (F.P.); 7Centre de Recherche des Cordeliers, Sorbonne Université, Université Paris Cité, 75006 Paris, France; 8Equipe Labellisée Ligue Contre le Cancer, 75006 Paris, France; 9Inovarion, 75005 Paris, France; 10Department of Pathology, Weill Cornell Medicine, New York, NY 10021, USA; szd3005@med.cornell.edu; 11Department of Pathology, University of California, San Francisco, CA 94143, USA; won-tak.choi@ucsf.edu; 12Institute of Pathology, First Faculty of Medicine, Charles University, General University Hospital in Prague, 12808 Prague, Czech Republic; pavel.dundr@vfn.cz; 13Department of Pathology, Istituto Nazionale Tumori IRCCS Fondazione G. Pascale, 80131 Napoli, Italy; f.tatangelo@istitutotumori.na.it (F.T.); annabella.dimauro@istitutotumori.na.it (A.D.M.); 14Department of Pathology, Cliniques Universitaires St-Luc, Institut de Recherche Clinique et Experimentale (Pole GAEN), Université Catholique de Louvain, 1348 Brussels, Belgium; pamela.baldin@uclouvain.be; 15Immunomonitoring Platform, Laboratory of Immunology, AP-HP, Assistance Publique-Hopitaux de Paris, Georges Pompidou European Hospital, 75015 Paris, France; 16Department of Pathology, AP-HP, Assistance Publique-Hopitaux de Paris, Georges Pompidou European Hospital, 75015 Paris, France; 17Digestive Surgery Department, AP-HP, Assistance Publique-Hopitaux de Paris, Georges Pompidou European Hospital, 75015 Paris, France; 18Institut Roi Albert II, Department of Medical Oncology, Cliniques Universitaires St-Luc, Institut de Recherche Clinique et Experimentale (Pole MIRO), Université Catholique de Louvain, 1030 Brussels, Belgium; marc.vandeneynde@uclouvain.be

**Keywords:** immunoscore, digital pathology, colon cancer, tumor microenvironment, prognostic markers, risk stratification, T cell, anatomopathology

## Abstract

**Simple Summary:**

This study aims to compare the performance of the standardized consensus Immunoscore (IS) digital pathology assay to an evaluation of the immune response via visual examination of hematoxylin–eosin (H&E) slides and CD3+/CD8+ stained slides, achieved by expert pathologists. Herein, we report the evaluation of 540 stained images by multi-institutional pathologists to determine the concordance between pathologist assessment before and after training. The results show that the IS assay outperformed expert pathologists’ T-score evaluation in the clinical setting. This reveals the potential of the IS as an immune pathology tool, critical for reproducible quantitative analysis of tumor-infiltrated immune cells. These findings can contribute to a better diagnosis, allowing one to stratify cancer patients into reliable prognostic groups, based on the immune parameters quantified by IS. This work will likely impact the management of colon cancer patients as it raises the importance of the implementation of digital pathology in cancer diagnosis to provide appropriate personalized therapeutic decisions.

**Abstract:**

Background: The Immunoscore (IS) is a quantitative digital pathology assay that evaluates the immune response in cancer patients. This study reports on the reproducibility of pathologists’ visual assessment of CD3+- and CD8+-stained colon tumors, compared to IS quantification. Methods: An international group of expert pathologists evaluated 540 images from 270 randomly selected colon cancer (CC) cases. Concordance between pathologists’ T-score, corresponding hematoxylin–eosin (H&E) slides, and the digital IS was evaluated for two- and three-category IS. Results: Non-concordant T-scores were reported in more than 92% of cases. Disagreement between semi-quantitative visual assessment of T-score and the reference IS was observed in 91% and 96% of cases before and after training, respectively. Statistical analyses showed that the concordance index between pathologists and the digital IS was weak in two- and three-category IS, respectively. After training, 42% of cases had a change in T-score, but no improvement was observed with a Kappa of 0.465 and 0.374. For the 20% of patients around the cut points, no concordance was observed between pathologists and digital pathology analysis in both two- and three-category IS, before or after training (all Kappa < 0.12). Conclusions: The standardized IS assay outperformed expert pathologists’ T-score evaluation in the clinical setting. This study demonstrates that digital pathology, in particular digital IS, represents a novel generation of immune pathology tools for reproducible and quantitative assessment of tumor-infiltrated immune cell subtypes.

## 1. Introduction

The AJCC/UICC-TNM classification system based on anatomic pathology evaluation of tumors provides useful yet incomplete prognostic information [1]. New ways to classify cancer focusing on tumor cells have only shown modest prediction accuracy and limited clinical usefulness [1,2]. However, an extensive literature review demonstrated a favorable prognostic impact of the pre-existing adaptive immune cells infiltrating tumors [1,3,4,5,6,7,8,9,10,11,12]. In colorectal cancer (CRC), we showed a correlation between the in situ densities of adaptive immune cells at the center of the tumor (CT) and the invasive margin (IM) with patients’ survival [3,8,12,13,14]. A meta-analysis of the literature revealed the prognostic value of immune cells and that cytotoxic CD8+ T-cell enrichment was associated with a good prognosis in 97% of the studies [15]. We showed that cytotoxic and memory T cells were predictive of clinical outcome in early-stage CRC (I/II). We further showed that histopathologic-based prognostic factors of CRC are associated with the state of the local immune reaction [8]. The assessment of CD8+ cytotoxic T lymphocytes in combined tumor regions provided an indicator of tumor recurrence beyond that of the AJCC/UICC-TNM staging [16,17,18]. This immune response was defined by the “Immunoscore” (IS) [15,19,20,21].

An international IS consortium quantified the pre-existing immunity on stage I/II/III CC patients by using the first worldwide recognized and standardized consensus IS assay. The results established the consensus IS as a powerful and robust immune classifier to predict patient’s prognosis [22]. A meta-analysis on more than 10,000 CC patients confirmed that the consensus IS provided a reliable estimate of the recurrence risk [23]. Its clinical utility was further reinforced by publications demonstrating the prognosis value of IS in four independent cohorts of stage III CC patients, including two randomized phase 3 clinical trials [24,25], and its predictive value in response to chemotherapy [24,26]. The clinical utility of IS in Stage II CC patients was validated in multiple cohorts [14,22,27,28,29,30,31,32]. The immune response measured with the consensus IS was introduced as essential and desirable diagnostic criteria for CRC in the latest (5th) edition of the WHO Digestive System Tumors classification. Moreover, IS was introduced in the 2020 European and 2021 Pan-Asian adapted European Organization for Medical Oncology (ESMO) Clinical Practice Guidelines for gastrointestinal cancers to refine the prognosis and, thus, adjust the chemotherapy decision-making process [33,34]. Therefore, it is of the utmost importance to compare the performance of the standardized IS consensus performed with digital pathology to an evaluation of the immune response through visual examination of hematoxylin–eosin (H&E) slides or via a visual examination of CD3+- and CD8+-stained slides by expert pathologists.

## 2. Materials and Methods

### 2.1. Immunostaining Evaluation

An international group of 10 expert gastrointestinal (GI) pathologists, half from the USA and half from Europe and Japan, evaluated stained CD3+ and CD8+ slides (*n =* 540) from 270 randomly selected full resections of CC cases (cohort demographic distribution and characteristics are presented in Appendix A). Pathologists performed a semi-quantitative visual assessment (T-score) of CD3+ and CD8+ and reported results for all cases blinded from IS results. Each pathologist evaluated the same 270 cases, before training (unsupervised evaluation) and after training (supervised evaluation). Pathologists’ visual assessment and training were conducted according to previously described methods [35]. Before training, all pathologists reported CD3+ staining, CD8+ staining and the overall T-score of each patient into 3 categories (High, Intermediate or Low). All pathologists had the same reference slides (*n =* 12), representing cases with known IS (High, Intermediate or Low). Images with CD3 and CD8 densities corresponding to High, Intermediate and Low cut points in CT and IM regions were provided. For the supervised evaluation, training of the pathologists was performed by providing 12 cases at the cutoff values for IS. Then, all pathologists reported their results accounting for several parameters into three categories (High, Intermediate or Low): CD3+ cell density in CT and IM of the tumor, overall CD3+ cell density, CD8+ cell density in CT and IM, overall CD8+ cell density and overall T-score for each patient. The concordance or discordance for the 10 independent T-score evaluations on the 270 cases and the concordance with the IS were evaluated for two (High, Low) and three categories (High, Intermediate, Low).

### 2.2. Immune Cell Infiltration Evaluation on H&E Slides

H&E slide evaluation for tumor-infiltrating lymphocytes (TILs) was performed by 11 independent evaluators on the same 270 representative CC cases. Each evaluation was performed on the same 270 cases, and each evaluator had the same reference slides (3 representative H&E slides for each IS category). The 11 independent evaluations of TIL were performed in CT and IM regions separately, and the TIL categorization was reported into two- and three-category IS for each case. The concordance or discordance for the 11 independent evaluations of TIL on 270 H&E slides and the concordance with the IS were evaluated for two- and three-category IS.

### 2.3. Immunohistochemistry

For each patient, a pathologist selected a tumor block containing CT and IM regions. Two consecutive tissue paraffin sections of 4 µm were processed for single immunohistochemistry staining with CD3 and CD8 antibodies, followed by DAB substrate (3,3′-diaminobenzidine) in the presence of peroxidase (HRP) enzyme, according to a previously described protocol [22]. Digital slides were obtained with a 20× magnification and a resolution of 0.45 µm/pixel.

### 2.4. Image Analysis

The stained CD3 and CD8 cell densities were determined in CT and IM regions using a specially developed IS^®^ analyzer software (INSERM/Veracyte, Marseille, France). The mean and the distribution of the staining intensities were monitored, providing an internal quality control of each slide.

### 2.5. IS Determination

For each case, CD3 and CD8 densities in CT and IM regions were converted into percentiles, as previously described [22]. The mean of the four percentiles obtained (two markers, two regions) was calculated and translated into the IS scoring system. IS categories were previously defined independently of clinical data [22]. These pre-defined categories were used herein: mean percentiles 0–25%, >25–70%, and >70–100% for IS Low, Intermediate and High, respectively. Additional analyses were performed with the pre-defined two-category IS: Low (0–25%) and Intermediate + High (25–100%). Repeatability Evaluation of IS method was performed according to previously described protocols [22,35].

### 2.6. Statistics

Statistical analysis was used to explore the following types of concordance: between individual pathologist’s T-score assessment (from CD3 and CD8 staining) and IS for all cases (*n* = 270), for the subset of cases around the clinical Low (25th percentile) IS cut point (*n* = 54) and for the subset of cases around the High (70th percentile) IS cut point (*n* = 54), before and after training, inter-pathologist agreement with visual assessment of T-score, among three repeated IS quantifications (*n* = 50) and between 11 visual evaluations of TIL (from H&E slides) and IS for all cases (*n* = 270). The Cohen’s Kappa coefficient was used to evaluate agreement of IS results between the two rating methods, IS and pathologists’ T-score, and between IS and TIL (H&E evaluation). The Fleiss’s Kappa coefficient test, an extension of the Cohen’s Kappa, was used to compute the agreement between multiple observers’ assessment. In accordance with McHugh [36], the level of agreement was categorized according to the Kappa values as: none (0–0.20), minimal (0.21–0.39), weak (0.40–0.59), moderate (0.60–0.79), strong (0.80–0.90) and almost perfect (>0.90%). A negative Kappa indicated that there was less agreement than would be expected by chance given the marginal distributions of ratings. Observers Needed to Evaluate Subjective Tests (ONEST) analysis was used to visualize the change in overall percent agreement as a function of the number of observers, as previously described [37]. High discordance amongst observers is found when the plateau begins at a higher number of observers and occurs at a low overall percent agreement. Ethical, legal and social implications were approved by an ethical review board from île de France (#0912082).

## 3. Results

CC samples from 270 randomly selected representative cases were stained for CD3 and CD8, with the consensus IS being computed (Appendix A). The consensus IS was established using the published pre-defined cut points [22] to convert CD3 and CD8 immune densities into percentiles and IS categories (Low, Intermediate, High). IS Low, Intermediate and High represented 33%, 49% and 18% of the cohort, respectively (Appendix A).

CD3- and CD8-stained slides (540 images) were given to 10 pathologists blinded to the IS results (Appendix A). Each pathologist evaluated the CD3+ and CD8+ cells on the whole slide (unsupervised analysis). The 10 independent evaluations of stained slides were performed in CT and IM regions separately. CD3 and CD8 stains were reported for each patient according to the pathologist’s visual expertise into three T-score categories (Low, Intermediate or High). Pathologists were then trained with 12 reference images of known IS values at the IS cut points (see Methods for details). After training, the pathologists re-evaluated the 540 images of CD3 and CD8 stains and reported their semi-quantitative T-scores once again for each patient (Figure 1A, Appendix A). Concordance between pathologists and concordance between pathologists and the consensus IS obtained with digital pathology were then analyzed.

### 3.1. Disagreement between Pathologists’ Visual Evaluation of CD3- and CD8-Stained Slides

Concordance between pathologists’ evaluation of CD3- and CD8-stained slides was analyzed (Figure 1B,C). Pathologists’ disagreement was defined as the percentage of non-concordant cases for which at least one pathologist’s assessment was different from others. Disagreement was observed in the vast majority of cases before training (94%) and after training (95%). Indeed, concordance between all pathologists was found for less than 9% and 4% of patients before (Figure 1B) or after training (Figure 1C), respectively. Discordance (one to four pathologists not agreeing with others) was observed in 54% and 45% of cases before and after training, while a random T-score classification (five pathologists with one T-score and five pathologists with another T-score) was found in 11% and 10% of patient samples before and after training, respectively. Strikingly, 26% and 41% of patient samples before and after training, respectively, were very discordantly scored by pathologists, with the same case being classified as High, Intermediate and Low. Of note, a detailed analysis among IS Low, Intermediate, and High categories revealed that IS Intermediate was the least concordant (Appendix A). Overall, this suggests that training had no effect on the pathologist’s ability to properly classify cases, independently of IS categories (Figure 1B,C).

### 3.2. Disagreement between Pathologists’ Visual Evaluation of CD3- and CD8-Stained Slides and IS Digital Pathology

We then aimed to compare concordance and discordance between pathologists and IS digital pathology analyses. For this matter, all cases were sorted from lowest IS (in blue) to highest IS (in red), before training (Figure 2A) and after training (Figure 3A). Heatmaps revealed a trend for a correlation between each pathologist’s evaluation and IS quantification (from left to right). However, heatmaps also revealed major discrepancies between pathologists (from top to bottom). Indeed, each case was classified as concordant, discordant, random or very discordant. Without training, pathologists’ disagreement was observed in the vast majority of the cases compared to digital pathology IS quantification (Figure 2B). After training, similar results were obtained with 95.5% of non-concordant cases (Figure 3B). In fact, concordance between all pathologists was only found in 8.6% and 4.5% of cases before and after training, respectively. Discordance was found in 54.1% and 44.8% of patients, while a random T-score classification was found in 11.6% and 10.0% of patients before and after training, respectively. Moreover, 26.5% and 40.7% of patients before and after training, respectively, were very discordantly scored by pathologists (Figure 2B and Figure 3B).

When evaluating concordance within each IS group (Low, Intermediate or High), concordant cases were mostly seen within the 5% lower and higher end of the Low and High IS categories, whereas non-concordance was observed across all categories. Indeed, discordant cases were spread across a large spectrum of IS (from 5% to 95% percentile) before training, whereas they were vastly associated with IS Low and High groups after training (Figure 2B and Figure 3B). On the other hand, very discordant cases were found for a broad range of IS, both before (Min–Max 6–84%, median 55% percentile) and after training (Min–Max 6–88%, median 41% percentile) (Figure 2B and Figure 3B).

Overall, these data suggest that pathologists are accurate in categorizing patients amongst the 5% with the lowest and highest T-cell infiltration (Low, High IS) but are not accurate enough for the rest, leaving behind the vast majority of patients (92.2% and 95.5% before and after training, respectively) (Figure 2B and Figure 3B).

### 3.3. Concordance between Pathologists’ T-Score Evaluation and Consensus IS Using Digital Pathology

The agreement between pathologists’ classification and the two- or three-category IS was evaluated via Cohen’s Kappa statistical analysis (Table 1). Without previous training, the agreements between pathologists’ evaluation for CD3 and CD8 staining classification and the reference IS assessment of 270 CC cases were weak (Table 1). Indeed, the mean Cohen’s Kappa was 0.498 (minimum and maximum agreements were (0.32, 0.59)) for the two-category IS (Low, High) and 0.408 (0.27, 0.52) for the three-category IS (Low, Intermediate or High). Similarly, after training, data showed a mean Kappa of 0.465 (0.282, 0.642) for the two-category IS and 0.374 (0.005, 0.566) for the three-category IS.

Furthermore, analysis of the 20% of CC cases around the IS clinical cut points (25%-Low, 70%-High) resulted in even lower concordance, with overall disagreement rates over 99%, both before and after training. This suggested that the pathologists’ patient classification did not improve after training (Table 1; Figure 2B and Figure 3B). Before training, Cohen’s Kappa index for all pathologists versus IS for the 20% of cases around the IS cut points revealed no concordance with the mean Kappa in two categories of 0.10 (min/max −0.12/0.33). Similar results, showing no concordance, were observed for both cut points (25%: Low, 70%: High), when grouping into two or three categories, before and after training (Table 1). We also analyzed the mean Cohen’s Kappa index for all pathologists and the Cohen’s Kappa index for each pathologist within subgroups of CC patients. The pathologist’s T-score classification and its concordance with IS were evaluated for T1–T2, T3, T4 and T3–T4 subgroups, for patients with or without mucinous colloid type and for different grades of tumor differentiation. All mean Cohen’s Kappa index values ranged between minimal and weak concordance (K = 0.21–0.59) (Appendix A).

Interestingly, pathologists would change their classification in 41.8% of cases after training, with a mean percentage gain of cases correctly classified averaging −4.2% (worse after training than before) (Figure 4A). This highlights the fact that pathologists often changed categories but were still inaccurate in the categorization of patients after training.

### 3.4. Comparison of Individual Pathologist’s Supervised Visual Assessment after Training to IS

The overall agreement rate was defined by the mean percentage of cases for which all pathologists’ evaluations were in accordance with the reference IS for the 270 CC patients. Only 8.6% and 4.5% of cases were concordant with the IS before and after training, respectively. Representative images of CD3 staining evaluated by a pathologist compared to the IS are illustrated together with clinical information, (T, N, M, MSI, number of lymph nodes, recurrence and death), including the whole tumor at low magnification and six high-magnification fields (Appendix A). Examples of patients with very discordant evaluation by pathologists (Appendix A–C), one extreme case of Low IS (IS = 2.5%) with concordant evaluation (Appendix A), and one extreme case of High IS (IS = 97.5%) with concordant evaluation (Appendix A) are provided.

The mean percentage of cases concordant before and after training with IS for each pathologist was only 41.8% (Type 1) (Figure 4A). The average proportion of Type 2 disagreement (discordant classification before and after training) was 19, and a high disagreement rate between the pathologists’ evaluation and the reference IS was observed before (37% of disagreement) and after (41% of disagreement) training (Figure 4B). After training, no gain in agreement was observed, but many cases (22%) correctly reported before training were reported incorrectly after training (Figure 4B). Indeed, 18% of cases were concordant after but not before training, but 22% of cases concordant before training were not concordant any longer after training.

### 3.5. Comparison of TIL Evaluation on H&E Slides to T-Score Evaluation and to Digital IS

Target plots illustrated the proportion of evaluation with concordance, discordance around cut points, discordance, random cases and very discordant cases for IS quantification (Figure 5A), TIL evaluation on H&E slides (Figure 5B), pathologists’ evaluation of CD3 and CD8 stains before training (Figure 5C) and pathologists’ evaluation of CD3 and CD8 stains after training (Figure 5D). TIL evaluation on H&E (Low, Intermediate or High) showed only 4% concordance between 11 evaluators, 51% of discordant cases and 45% of very discordant non-conclusive cases. IS quantification using digital immune pathology was more reproducible than visual evaluation of H&E slides or CD3+- and CD8+-stained slides.

ONEST analysis was used to determine the minimum number of evaluators needed to estimate concordance between several readers [37]. ONEST plots showed decreasing overall percent agreement as the number of observers increased, reaching a low plateau of 0.25 at ten observers for T-score in two categories and of <0.1 agreement for T-score in three categories (Appendix A).

Finally, target plots illustrated almost perfect concordance (Cohen’s Kappa K > 0.9) in the reproducibility of IS, in two- or three-category IS, using digital pathology quantification (Figure 5). In contrast, no concordance (Cohen’s Kappa K < 0.25) was observed between TIL evaluated on H&E slides and IS. A weak or minimal concordance (Cohen’s Kappa K < 0.5) was observed between pathologists’ visual evaluation of stained CD3 and CD8 slides, both before and after training and with known IS cases. No concordance (Cohen’s Kappa K < 0.12) was observed between pathologists’ visual evaluation of stained CD3 and CD8 slides, both before and after training and with the 20% of cases around the IS cut-point categories (Appendix A). The clinical utility of IS is illustrated with a treatment and surveillance TNM-IS decision tree. In stage II, IS High with low to no recurrence, clinicians could consider surgical resection only and low-intensity surveillance, in contrast to IS-Low patients. Overall, IS could impact treatment decision making between 23 and 48% and could impact surveillance decision making for 48% of the patients with stage II colon cancer (Appendix A). In stage III, IS could impact treatment decision making for 55% of the patients with stage III colon cancer. Visual evaluation of T-score by a pathologist would lead to 70% of cases being non-concordant, leading to inapropriate treatment and surveillance (Appendix A).

## 4. Discussion

### 4.1. Reproducibility of IS (Specificity, Sensibility, Kappa and Concordance)

Multiple analyses and meta-analyses have highlighted the role of T lymphocytes and cytotoxic T cells having a major influence on patient survival [15,18,20,21,38,39,40,41]. The immune response, as measured by IS, was introduced for the first time in the latest 5th edition of the WHO Digestive System Tumors as “essential and desirable diagnostic criteria for colorectal cancer”. In addition, IS was introduced in the 2020 European and 2021 Pan-Asian adapted European Organization for Medical Oncology (ESMO) Clinical Practice Guidelines for gastrointestinal cancer to refine the prognosis and to adjust the chemotherapy decision-making process [33,34]. As previously documented, analytical validations of IS highlighted that it is a robust, reproducible, quantitative and standardized immune assay, with a high prognostic performance, independent of all the prognostic markers currently used in clinical practice [42]. IS percentile values remained remarkably constant between formalin-fixed paraffin-embedded tissue blocks from the same patient. The correlation coefficients were R = 0.94 and R = 0.97 for CD8 and CD3, respectively. The concordance between results obtained with the selected blocks and the random blocks was 93% (95% CI 88–96%) [42]. The reproducibility of IS was evaluated on 13 slides per block for 10 patients and revealed excellent accuracy (95.7%), sensitivity (94.8%), specificity (100%) and an overall ROC area of 0.99 [42]. The technical variability of the method was evaluated with lot-to-lot reproducibility and IS assay precision measurements. Consecutive slides from three CCs were assessed for CD3+ and CD8+ T-cell densities using three different antibody lots, three DAB revelation kit lots, two different benchmark auto-stainers, three different runs and three different operators. A concordance of 100% was observed between IS categories [42]. The analytical variability of the quantification by digital pathology was evaluated. Representative cases (*n =* 36) with ISs ranging from 2.5th to 90th percentiles were re-analyzed by eight independent pathologists from different centers. Mean cell densities for CD3 and CD8 in each tumor region revealed a strong inter-observer reproducibility (r = 0.97 for tumor; r = 0.97 for invasive margin; *p* < 0.0001) [22]. A full assessment of IS reproducibility was performed in two laboratories. Each laboratory had its own IS workflow, including staining, scanning and analysis. Non-consecutive cutting slides from the same tumor block were used to assess the IS of 100 representative cases. The inter-laboratory correlation for CD3+ and CD8+ cells was 0.94 (*p* < 0.001), and the overall categorical IS concordance between the two centers was 93%. This also included biological variability of the tumor [42]. Moreover, the rare cases of discordance were all very close to the cut-point value of 25%, and it would be easy to re-test IS in such samples to correctly assign their score. Finally, the concordance from five independent IS quantifications using Cohen’s Kappa statistics revealed an almost perfect concordance (K > 0.93) between digital quantifications of IS [35].

### 4.2. Non-Reproducibility of TIL on H&E Slides

A visual assessment of the density of TILs in tumor tissue stained with H&E was analyzed. H&E images from representative cases (*n =* 270) from the international SITC cohort were assessed by 11 observers. Only 4% of cases were concordant between all observers, 8% of cases were concordant between 80% of observers and a total absence of concordance (50% discordance) was evident in 45% of the cases [22].

Concordance between all observers was obtained for only 8% of cases and was concordant with the digital IS for only 3% of cases. Discordant cases, with at least one evaluation different from others and different from IS, were found in 25% of cases. Strikingly, very discordant cases (the same H&E slide being evaluated as Low, Intermediate or High) were found in 72% of cases. The difference between IS quantification and TIL evaluation on H&E slides not only reflects the difficulty of such evaluation but also indicates that H&E staining of TILs is a crude and subjective semi-quantitative evaluation of undefined cell populations with possible opposite functions, such as CD4+ T cells with Th1 orientation vs. Th2 orientation vs. immune cells with regulatory functions (Treg cells), natural killer (NK) cells, NK-T cells, B-cells, subsets, innate lymphoid cells, cytotoxic CD8 T cells, or even round-shaped monocytes. This illustrates the complexity, subjectivity and discordance of TIL evaluation on H&E slides.

### 4.3. Non-Concordance between Pathologists for T-Score Evaluation before and after Training

The semi-quantitative evaluation of 540 chromogenic (DAB) single-stain slides (CD3+ and CD8+) by 10 pathologists revealed major discordance between pathologists. Indeed, before training, evaluation showed 91% of non-concordant cases between pathologists. Furthermore, 26% of cases were very discordant (the same slides from the same patient being evaluated Low, Intermediate or High by different pathologists). These discrepancies were not improved after training with 12 representative reference CD3 and CD8 cases at the IS cut points (25th and 70th percentile).

A significant disagreement was observed between the semi-quantitative pathologist’s T-score (into two (High or Low) or three categories (High, Intermediate, Low)) compared to the consensus digital pathology IS. Importantly, a high rate of disagreement was observed when comparing the pathologists’ visual assessment with the reference IS, leading to misclassification of >96% cases, and this disagreement was even higher (100%) for the cases around the clinical cut point (of 25th percentile). The study revealed that the impact of training was heterogeneous between pathologists and that, overall, training did not improve the concordance between the visual assessment and IS. Changes in training methods could be considered; however, this also illustrates the complexity, subjectivity and discordance of CD3+ and CD8+ evaluation by visual examination.

The lack of improvement in agreement between pathologists’ evaluation and quantitative digital pathology, before and after training, is likely multifactorial. In fact, the size of a colon tumor is quite large, and a whole slide analysis revealed a heterogeneous pattern of CD3+ and CD8+ within different areas of the tumor. The total number of CD3+ cells on a given slide (CT + IM regions) is huge, with a mean of 88,000 CD3+ T-cell/slide, making visual evaluation very challenging. Furthermore, the mean density of these cells is higher at the invasive margin compared to the core of the tumor, rendering the overall visual evaluation difficult. In addition, these immune cells can be present at different densities within the tumor or the stroma and can be clustered or dispersed, even within the same tumor. CD3+, encompassing both CD8+ and CD4+ T-helper cells, and CD8+ cells also have different densities in different areas of the tumor, and the evaluation has to be performed twice for each of these markers on consecutive slides. Looking at the overall slide is tedious, and the semi-quantitative evaluation of so much heterogeneity is very complex and, in fact, very subjective. It is likely that poor concordance would also have been observed within pathological subgroups. The poor performance of pathologists’ scoring even after training demonstrated that the novel tool of quantitative digital immune pathology is clearly much more appropriate for such evaluations. Even in an easier context of PD-L1 in non-small cell lung cancer (NSCLC), there is an impact of a pathologist’s personality on the interobserver variability and diagnostic accuracy of immunostaining [43]. Furthermore, the subjective T-score showed low reproducibility across multiple pathologists with ONEST analysis, suggesting that the vast majority of pathologists will disagree about subjective evaluation of infiltrating T cells. Thus, based on previous data [22,35] and our actual results obtained herein, the digital consensus IS quantification shows a high level of reproducibility, with perfect software concordance. In contrast, the pathologists’ visual subjective evaluation on H&E slides or the evaluation of CD3+- and CD8+-stained slides was not reliable enough for a precise therapeutic decision-making process. In conclusion, a reliable evaluation of CD3+ and CD8+ cells and of IS on a whole slide section shall not be a visual estimation but rather a real IS quantification using the dedicated reproducible software.

### 4.4. Clinical Impact of Misclassification

The pre-existing immune contexture has an impact on the response to chemotherapy and immunotherapy treatments [16,20,44,45,46,47,48,49,50,51,52,53,54,55]. Multiple therapeutic approaches against cancer are ongoing [15,39,46,47,49,53,56,57,58,59], and for quantitative immune classification and the precision management of patients, biomarkers using quantitative pathology are becoming a necessity [42,60]. Misclassification of stage II and III CC patients by T-score semi-quantitative evaluation would result in inappropriate treatment decision making for many patients [24,25,26]. Similarly, another assay, Immunoscore-IC, which is also a quantitative and spatial evaluation of immune markers (CD8 and PD-L1), predicts response to immunotherapy and requires digital pathology [61,62].

For patients with stage II CC, many patients being misidentified as stage II CC at low clinical risk would, in fact, be at high risk based on IS. Such a situation would produce false expectations of recurrence for these patients who will not be monitored as closely as those at high risk of recurrence to detect signs of relapse earlier. These patients would not be appropriately considered as high-risk stage II patients and may be under-surveilled and under-treated. Similarly, misclassification of truly IS-High stage II CC patients as having tumors with low T-score by visual examination could result in patients recommended for adjuvant chemotherapy when their recurrence risk is low and exposing them to unnecessary toxicity.

For stage III CC, IS-Low cases being misclassified as high with visual T-score would be detrimental, as patients who may not get benefit from chemotherapy [26] or longer duration of adjuvant chemotherapy (6 months versus 3 months) [24] would not be identified as poor responders. These patients may be unnecessarily subjected to additional chemotherapy and its associated long-term toxicity. Finally, stage III CC patients with IS-High could be misclassified as patients with poorly infiltrated tumors with visual evaluation (low T-score) and would not be identified as deriving significant benefit from a longer duration of adjuvant chemotherapy [24]. Such patients would be under-treated and subjected to an increased risk of relapse.

Based on previous treatment decision trees, IS would impact treatment decision for 44% of stage II patients and for 55% of stage III patients [63]. Based on visual T-scoring from a single pathologist after training, 70% of cases would be non-concordant with IS. Given an estimated incidence of 101,420 and 23,000 stage II and stage III CC patients per year, respectively, pathologists’ visual evaluation of T-score would lead to 70,914 stage II, 16,100 stage III and more than 87,000 CC cases being misclassified and possibly receiving inappropriate patient care annually.

## 5. Conclusions

The very important difference between pathologists’ T-score classification and the reproducible IS quantification highlights the importance of new tools for pathologists, namely quantitative digital pathology.

The potential negative impact of immune response misclassification due to pathologists’ T-score may result in erroneous prognosis and risk evaluation for many CC patients. These results demonstrated that the IS assay helps to better stratify patients into reliable prognostic recurrence groups. We conclude that the standardized and robust IS assay outperforms the assessment of expert pathologists in the clinical setting for immune response and can, thus, provide most appropriate individualized therapeutic decisions for CC patients.

## 6. Patents

J.G. and B.M. have patents associated with immune prognostic biomarkers. J.G. is co-founder of HalioDx, a Veracyte company. Immunoscore^®^ is a registered trademark owned by the National Institute of Health and Medical Research (INSERM) and licensed to Veracyte.

## Figures and Tables

**Figure 1 cancers-15-04045-f001:**
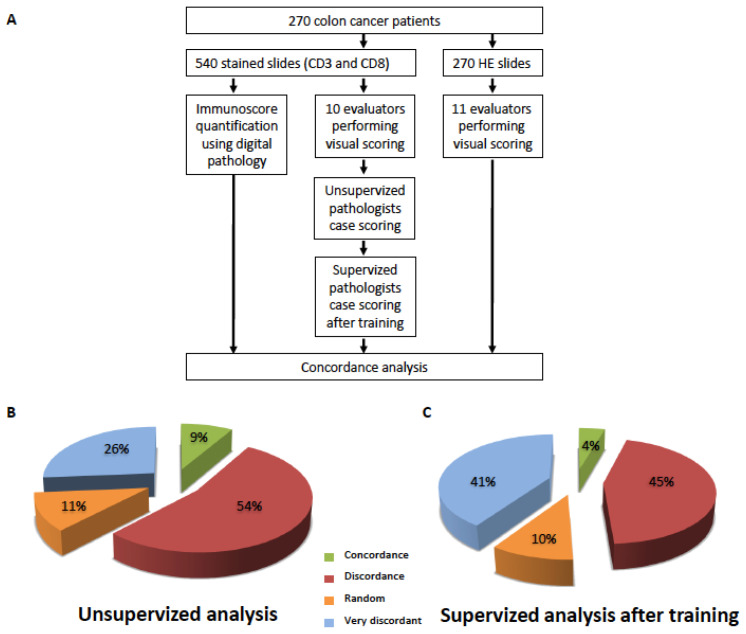
Schematic representation of the experimental design: 270 colon cancer patients from the international SITC cohort were selected for this study. For each patient, 2 consecutive whole slide samples were stained for CD3 and CD8 and 1 whole slide sample was stained using hematoxylin–eosin (H&E). CD3+ and CD8+ T cells of those stained slides were analyzed via either digital pathology (IS) or visual assessment by ten pathologists (T-score) before and after supervised training. In parallel, eleven pathologists were given H&E slides to visually assess the density of tumor-infiltrating immune cells in tumor tissue stained with H&E. Concordance between pathologists’ T-score and the digital IS was evaluated for two- and three-category IS. The concordance between the evaluation of the tumor-infiltrating immune cells on the corresponding H&E slides and the digital IS was also analyzed (**A**). Concordance analysis between individual pathologist’s T-score assessment. Pathologist’s T-score was evaluated based on two- (Low, High) and three-category IS (Low, Intermediate and High). Semi-quantitative evaluation of whole slide images for CD3+ and CD8+ cells was performed by pathologists blinded to IS results, before and after training. Pathologists’ disagreement was defined as the percentage of non-concordant cases for which at least one pathologist assessment was different from others, before (**B**) and after (**C**) supervised training. Results fall into four concordance levels: concordant (all pathologists agreeing on scoring), discordant (1 to 4 pathologists not agreeing with others), very discordant (one patient being scored High, Intermediate or Low) and random (5 pathologists with a T-score and 5 pathologists with another T-cell score).

**Figure 2 cancers-15-04045-f002:**
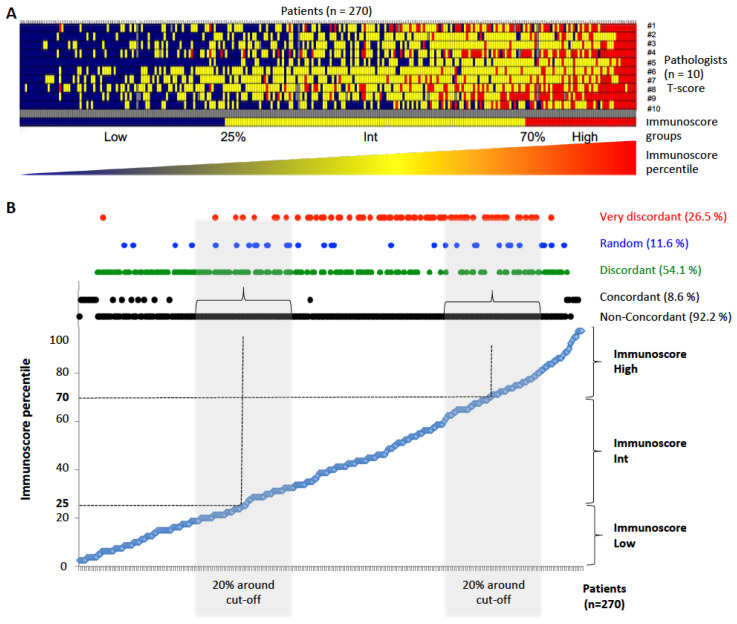
Concordance analysis between pathologists’ T-score and IS before training. (**A**) Heatmap representing plotted data for each pathologist blinded to digital pathology IS results before training. Ten pathologists (#1 to #10) evaluated the 270 patients to attribute their T-scores. Patients were illustrated from lowest (blue) to Intermediate (yellow), to highest (red) IS. (**B**) Graph displaying concordance for each IS percentile (<25% = IS Low. >70% = IS High. 25% < IS Intermediate < 70%) and for each concordance level before training: concordant, discordant, random and very discordant. Non-concordant results group everything bare concordant results; 20% around cut points was also used for the statistical Cohen’s Kappa results for concordance before training (cf. Table 1).

**Figure 3 cancers-15-04045-f003:**
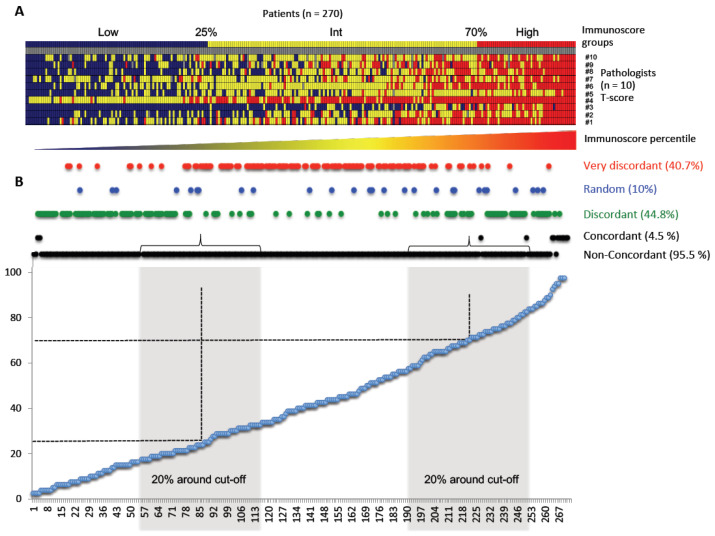
Concordance analysis between pathologists’ T-score and IS after training. (**A**) Heatmap representing plotted data for each pathologist blinded to digital pathology IS results after training. Ten pathologists (#1 to #10) evaluated the 270 patients to attribute their T-scores. Patients were illustrated from lowest (blue) to Intermediate (yellow), to highest (red) IS. (**B**) Graph gathering concordance for each IS percentile (<25% = IS Low. >70% = IS High. 25% < IS Intermediate < 70%) and for each concordance level after training: concordant, discordant, random and very discordant. Non-concordant results group everything bare concordant results; 20% around cut points was also used for the statistical Cohen’s Kappa results for concordance after training (cf. Table 1).

**Figure 4 cancers-15-04045-f004:**
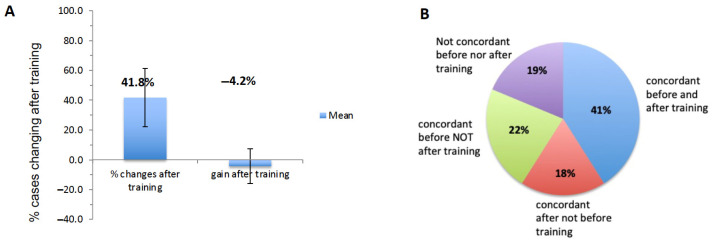
Mean changes of categories after training in pathologists’ supervised visual evaluation compared to IS. (**A**) The left histogram represents the mean percentage of changes in categories (Low, Intermediate, High) after training in supervised visual evaluation. The right histogram represents the mean percentage gain (+) or loss (−) of correctly classified cases after training. (**B**) Average proportions of concordance between pathologists and IS before and after training.

**Figure 5 cancers-15-04045-f005:**
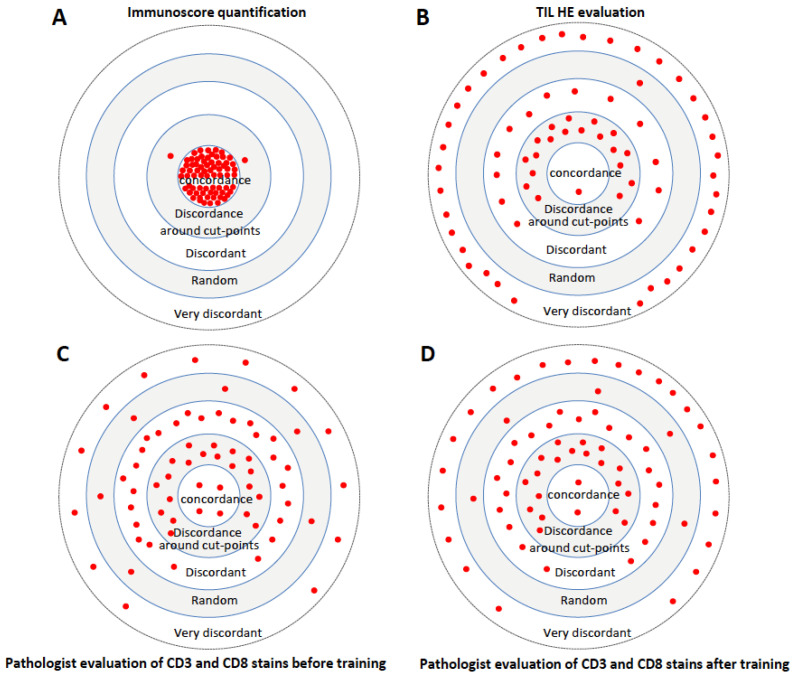
Target plot visualizations of concordance between pathologists’ evaluation of T-score and IS, before and after training. Proportion of evaluation with concordant, discordant around cut points, discordant, very discordant and random cases for IS quantification (**A**), immune-infiltrating lymphocyte evaluation on hematoxylin–eosin (H&E) slides (**B**), pathologists’ evaluation of CD3 and CD8 stains before training (**C**) and pathologists’ evaluation of CD3 and CD8 stains after training (**D**). Each dot illustrates 4 patients.

**Table 1 cancers-15-04045-t001:** Cohen’s Kappa statistical analysis highlighting agreements between pathologists’ T-score and the reference IS for 270 colon cancer patients, before and after training. Each comparison was performed for the pathologist’s classification (T-score) versus IS for the same sample and measured via Cohen’s Kappa for all 270 patients and for patients around the 20% cut-points Low (*n* = 54) and High (*n* = 54) (cf. Figure 2 and Figure 3). ** Kappa: worse than random (negative Kappa scores), none (0–0.2), weak (0.4–0.59), moderate (0.6–0.79), strong (0.8–0.9) and almost perfect (>0.9).

	Lo vs. Int vs. Hi */Classification (3 Groups) **				Lo vs. Int + Hi */Classification (2 Groups) **			
	Supervised		Unsupervised		Supervised		Unsupervised	
All patients (270 pts)								
Pathologist	Kappa	Concordance	Kappa	Concordance	Kappa	Concordance	Kappa	Concordance
1	0.350	minimal	0.378	minimal	0.486	weak	0.471	weak
2	0.413	weak	0.469	weak	0.550	weak	0.574	weak
3	0.361	minimal	0.394	minimal	0.391	minimal	0.496	weak
4	0.005	none	0.381	minimal	0.058	none	0.444	weak
5	0.448	weak	0.385	minimal	0.579	weak	0.508	weak
6	0.566	weak	0.461	weak	0.642	moderate	0.565	weak
7	0.258	minimal	0.396	minimal	0.282	minimal	0.486	weak
8	0.465	weak	0.421	weak	0.568	weak	0.517	weak
9	0.420	weak	0.526	weak	0.574	weak	0.593	weak
10	0.456	weak	0.270	minimal	0.520	weak	0.328	minimal
20% around 25% low (54 pts)								
Pathologist	Kappa	Concordance	Kappa	Concordance	Kappa	Concordance	Kappa	Concordance
1	0.001	none	0.054	none	0.051	none	0.038	none
2	0.002	none	0.189	none	0.024	none	0.189	none
3	0.092	none	0.146	none	0.092	none	0.146	none
4	−0.106	worse than random	−0.013	worse than random	0.000	none	−0.024	worse than random
5	0.238	minimal	0.152	none	0.262	minimal	0.152	none
6	0.329	minimal	0.071	none	0.329	minimal	0.071	none
7	0.020	none	−0.012	worse than random	−0.059	worse than random	0.008	none
8	0.089	none	0.316	minimal	0.112	none	0.329	minimal
9	0.040	none	0.156	none	0.091	none	0.203	none
10	0.167	none	−0.120	worse than random	0.151	none	−0.120	worse than random
20% around 70% high (54 pts)								
Pathologist	Kappa	Concordance	Kappa	Concordance				
1	0.186	none	0.057	none				
2	0.047	none	0.182	none				
3	0.149	none	0.037	none				
4	0.063	none	0.019	none				
5	0.191	none	0.123	none				
6	0.139	none	0.082	none				
7	−0.104	worse than random	0.177	none				
8	0.031	none	0.286	minimal				
9	0.157	none	0.199	none				
10	0.305	minimal	−0.026	worse than random				

* Each comparison is done for the pathologist’s classification vs. the Gold Standard Immunoscore for the same sample and measured by Cohen’s Kappa. ** Kappa: worse than random (negative Kappa), none (0–0.2), minimal (0.21–0.39), weak (0.4–0.59), moderate (0.6–0.79), strong (0.8–0.9), and almost perfect (>0.9).

## Data Availability

The materials described in the manuscript are freely available to use for non-commercial purposes. Detailed extracted data can be provided immediately following publication, upon request to the corresponding author. Proposals should be directed by email to the corresponding author J.G., at jerome.galon@crc.jussieu.fr.

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
