# Peer review of "Multi-Institutional Evaluation of Pathologists’ Assessment Compared to Immunoscore"

_cancers, 2023, doi:10.3390/cancers15164045_

Round 1

Reviewer 1 Report

In this study, the authors utilized the Immunoscore (IS) digital pathology assay to assess the T-score of CD3+ and CD8+ cells based on 540 images from 270 randomly selected colon cancer (CC) cases. Compared to expert pathologists, IS demonstrated superior reproducibility, including specificity, sensitivity, kappa, and concordance.

Previous studies have shown that IS can effectively predict prognosis, disease recurrence, and chemotherapy resistance in CC patients, making it a potentially valuable tool in clinical practice. However, this article only compared the concordance between IS and pathologists based on the aforementioned images, specifically the visual assessment of tumor-infiltrating lymphocytes (TILs) density in tumor tissues. The results showed only 3% concordance between observers and IS. The article did not thoroughly analyze how this superior performance of IS could improve diagnosis, treatment, and prognosis for patients. Therefore, it is suggested that the authors collect more clinical information to further enhance the practical implications of their findings.

Regarding the unsupervised and supervised evaluation, despite the potential benefits of factors such as repeated measurements and corrected methods before and after training, the concordance between pathologists and IS did not improve and even slightly decreased. The lack of improvement in agreement is surprising and raises questions about the effectiveness of the training method. This aspect of the results may mislead readers about the importance of training in clinical practice.

IS is a tool used to evaluate immune response in cancer patients and has shown advantages in measuring CD8+ T cells in CC pathology slides, including high sensitivity and specificity. Since CD8+ T cells are important indicators of immune checkpoint inhibitors (ICI) efficacy, combining IS with ICI response prediction could significantly enhance the clinical relevance of the article.

Furthermore, current research on immune response primarily focuses on predicting the efficacy of immune checkpoint inhibitors (ICIs). In patients with colon cancer, MSI-H has emerged as a potential biomarker for effective immunotherapy. However, existing studies indicate that MSI-H is not significantly correlated with PD-L1 expression in colon cancer, and the predictive role of PD-L1 in the efficacy of immune therapy for colon cancer is not ideal. Therefore, it is crucial to select appropriate tumor types to study the role of IS in evaluating immune response.

Overall, the clinical value of the article is insufficient, and the topic lacks novelty. 

Minor editing of English language required

Author Response

Dear Reviewer,

We thank you very much for reviewing our manuscript and for providing relevant comments that we have addressed point by point during our revisions.

Please find attached our revisions on your comments and suggestions.

Thank you for your consideration.

Best regards.

Jerome Galon.

Reviewer 2 Report

Manuscript entitled "Multi-institutional evaluation of pathologists’ assessment compared to Immunoscore"

Major issues:

1. The tables are not complete (due to the error in formating). It should be corrected.

2. The kappa value of the different pathologists should be listed to convince the readers.

3. Soem representative cases with excellent, good, and poor correlation should be illustrated in more detail.

4. The authors should also include some cases with known therapeutic response (outcome).

acceptable.

Author Response

(The authors gave the same response as above.)

Round 2

Reviewer 1 Report

The author has responded to each of the reviewer's comments in the revised manuscript, and provided additional clinical details about the patient cohort along with a more nuanced explanation of the T-score calculations used for comparison. The inclusion of the TNM-IS decision-tree adds valuable clarity regarding the potential clinical applications of the IS.

Furthermore, the author has expanded upon the initial IS findings by conducting an extensive series of follow-up studies. These new experiments reinforce the ability of IS to effectively predict ICI efficacy.

Taken together, IS seems to be a promising tool in clinical application. The quality of the manuscript has been substantially enhanced and deserves publication.

Minor editing of English language required

Reviewer 2 Report

The revision is acceptable

acceptable